# Bioprospecting Bioactive Peptides in *Halobatrachus didactylus* Body Mucus: From In Silico Insights to Essential In Vitro Validation

**DOI:** 10.3390/md23020082

**Published:** 2025-02-13

**Authors:** Marta Fernandez Cunha, Ezequiel R. Coscueta, María Emilia Brassesco, Frederico Almada, David Gonçalves, Maria Manuela Pintado

**Affiliations:** 1CBQF—Centro de Biotecnologia e Química Fina—Laboratório Associado, Escola Superior de Biotecnologia, Universidade Católica Portuguesa, Rua Diogo Botelho 1327, 4169-005 Porto, Portugal; mfcunha@ucp.pt (M.F.C.); mbrassesco@ucp.pt (M.E.B.); mpintado@ucp.pt (M.M.P.); 2MARE—Marine and Environmental Sciences Centre, ISPA Instituto Universitário de Ciências Psicológicas, Sociais e da Vida, Rua Jardim do Tabaco, 34, 1149-041 Lisbon, Portugal; 3Institute of Science and Environment, University of Saint Joseph, Rua de Londres 106, Macau SAR, China; david.goncalves@usj.edu.mo

**Keywords:** bioactive peptides, peptide aggregation, fish mucus, marine bioprospection

## Abstract

Fish body mucus plays a protective role, especially in *Halobatrachus didactylus*, which inhabits intertidal zones vulnerable to anthropogenic contaminants. In silico predicted bioactive peptides were identified in its body mucus, namely, EDNSELGQETPTLR (HdKTLR), DPPNPKNL (HdKNL), PAPPPPPP (HdPPP), VYPFPGPLPN (HdVLPN), and PFPGPLPN (HdLPN). These peptides were studied in vitro for bioactivities and aggregation behavior under different ionic strengths and pH values. Size exclusion chromatography revealed significant peptide aggregation at 344 mM and 700 mM ionic strengths at pH 7.0, decreasing at pH 3.0 and pH 5.0. Although none exhibited antimicrobial properties, they inhibited *Pseudomonas aeruginosa* biofilm formation. Notably, HdVLPN demonstrated potential antioxidant activity (ORAC: 1.560 μmol TE/μmol of peptide; ABTS: 1.755 μmol TE/μmol of peptide) as well as HdLPN (ORAC: 0.195 μmol TE/μmol of peptide; ABTS: 0.128 μmol TE/μmol of peptide). Antioxidant activity decreased at pH 5.0 and pH 3.0. Interactions between the peptides and mucus synergistically enhanced antioxidant effects. HdVLPN and HdLPN were non-toxic to Caco-2 and HaCaT cells at 100 μg of peptide/mL. HdPPP showed potential antihypertensive and antidiabetic effects, with IC_50_ values of 557 μg of peptide/mL for ACE inhibition and 1700 μg of peptide/mL for α-glucosidase inhibition. This study highlights the importance of validating peptide bioactivities in vitro, considering their native environment (mucus), and bioprospecting novel bioactive molecules while promoting species conservation.

## 1. Introduction

Bioprospection is the exploration of natural bioresources, namely, animals, plants, and microorganisms, whose potential may lead to the development of products that benefit society [1]. As a largely untapped reservoir of bioresources, the marine environment has become attractive for bioprospection due to the promising properties of new marine chemical compounds [2,3]. *Halobatrachus didactylus*, also known as Lusitanian Toadfish, though understudied as a potential source of bioactive compounds, possesses unique characteristics that highlight its potential for further investigation. This species secretes abundant mucus, and based on our observations during the collection of mucus samples in aquariums, fish kept in captivity alongside other species often caused a rapid death of these cohabiting fish. This suggests a potential toxic effect of substances released into the water, potentially originating from the mucus or other sources, such as axial glands [4].

In a previous publication, we unveiled the bioactive properties present in the body mucus of *H. didactylus*, including potential antioxidant, antimicrobial, and antihypertensive activities [4]. Additionally, we conducted size exclusion chromatography to obtain the protein–peptide profile and isolate the most prominent peak which contained the peptide fraction presumed to be responsible for the observed bioactivities. Subsequently, nano-liquid chromatography–tandem mass spectrometry (Nano-LC-MS/MS) determined the most probable peptide sequences associated with the peptide fraction. Bioactivity prediction databases, specifically, PeptideRanker (http://distilldeep.ucd.ie/PeptideRanker/), CAMPR_3_ (http://www.camp3.bicnirrh.res.in), and BIOPEP–UWM (https://biochemia.uwm.edu.pl/biopep/start_biopep.php), accessed on 15 January 2025, were employed to screen bioactive peptides. A deep analysis successfully identified five peptides with putative bioactivity: HdKTLR (EDNSELGQETPTLR), HdKNL (DPPNPKNL), HdPPP (PAPPPPPP), HdVLPN (VYPFPGPLPN), and HdLPN (PFPGPLPN). These bioactivities include inhibition activity against the following enzymes: dipeptidyl peptidase III (DPP-III), related to cancer and inflammation; dipeptidyl peptidase IV (DPP-IV), associated with type 2 diabetes; the angiotensin-converting enzyme (ACE), related to hypertension; and α-glucosidase, related to diabetes. Additionally, the peptides were predicted with promising antioxidant and antimicrobial properties.

The employment of an in silico approach involves computational methods that manipulate, interpret, and organize information within biological systems [5]. An in silico analysis significantly accelerates the discovery of new bioactive peptides by facilitating the prediction of bioactivities through computational modelling and machine learning. Databases such as BIOPEP gather essential information, including peptide sequences, sources of origin, reference articles, and documented bioactivities [6], providing a comprehensive resource for researchers. This approach proves highly effective in uncovering the bioactivities of peptides, offering high precision and efficiency while saving time, costs, and resources [7]. However, it is crucial to establish a strong correlation between experimental analyses (in vitro assays) and in silico methods [8,9]. This integrated approach enables substantial advancements in the discovery of bioactive peptides.

This study focused on exploring new bioactive molecules, particularly peptides, from the mucus of *H. didactylus*. This involved validating in silico predicted bioactivities through in vitro assays, which is crucial in determining the peptide bioactive potential. Additionally, it was important to study the conditions under which these peptides can achieve optimal bioactivity. Issues such as aggregation may lead to the reduced efficacy and stability of peptides, making it essential to address these challenges during the development process.

## 2. Results and Discussion

### 2.1. Peptide Aggregation and Stability Analysis

#### 2.1.1. Molecular Characterization of Aggregated Peptides

A high-performance size exclusion chromatography (HPSEC) analysis was employed to validate the molecular weight of the selected peptides, HdKTLR, HdKNL, HdVLPN, and HdLPN, synthesized by GenScript Biotech (Rijswijk, The Netherlands), following the preparation instructions provided in the technical datasheet. The decision to exclude HdPPP was based on its solubility just in pure dimethyl sulfoxide (DMSO), which presents challenges due to the high viscosity of the solution. At higher concentrations, viscosity may increase due to molecular interactions between DMSO, buffer ions, and water. In isocratic systems, such as HPSEC, an increase in the mobile phase viscosity can reduce flow efficiency, affecting resolution and increasing the system pressure. Additionally, DMSO can alter interactions between the mobile phase and the column matrix.

The HPSEC column was eluted with 344 mM of a phosphate buffer at pH 7.0, replicating the conditions used to analyze the mucus peptide fraction in our previous article [4]. During the analysis, we observed the formation of peptide aggregates, specifically oligomers, including dimers, trimers, and tetramers. Various buffer conditions were tested to determine if a particular pH and ionic strength would enable the peptides to expose their theoretical molecular weights, as specified in the GenScript technical datasheet. Since seawater typically has an ionic strength of approximately 700 mM [10], buffers with this ionic strength were tested at pH 3.0, 5.0, and 7.0, as well as at 344 mM of ionic strength.

An analysis of the chromatograms of the HdKTLR peptide under various buffer conditions revealed that the theoretical molecular weight of 1588.4 Da was not validated due to the formation of aggregates into oligomers. Notably, the buffer conditions most closely approximating the theoretical molecular weight was pH 3.0, with ionic strengths of 344 mM and 700 mM (Figure 1A). These findings also applied to the chromatograms of the HdKNL peptide, which has a theoretical molecular weight of 893.8 Da, with the observation that aggregate formation increased at higher pH conditions (Figure 1B). For the HdLPN and HdVLPN peptides with a theoretical molecular weight of 837.5 Da and 1099.6 Da, respectively, the degree of polymerization was less prominent than in the other peptides (Figure 1C,D), with HdLPN forming only dimers (Figure 1D).

Peptides prone to aggregation typically exhibit broader peaks in chromatograms, which is more prominently observed in the HdKTLR peptide. In size exclusion chromatography, increasing the salt concentration to enhance ionic strength reduces electrostatic interactions [11]. A review by Fekete et al. points out that at very high concentrations, these ions can promote hydrophobic interactions or ion-exclusion effects, leading to undesired secondary ionic interactions between peptides and the stationary phase, resulting in further aggregation. Additionally, adjusting the pH closer to the isoelectric point of the peptide can minimize secondary interactions. In our study, peptide aggregation occurs at both ionic strengths, with only pH adjustments causing the peptide’s molecular weight to approach the theoretical value.

Understanding that seawater typically has an ionic strength of around 700 mM is crucial, as fish are inherently adapted to their environmental conditions. Peptides exhibit conformational sensitivity due to environmental factors, such as seawater composition, specific cations, and anions in solution, and interactions with polyelectrolytes [12]. This highlights the sensitivity of peptide behavior to specific physicochemical parameters, underscoring the requirement for a comprehensive understanding of their conformational dynamics. Critical factors, including the molar extinction coefficient, water solubility, net charges at physiological pH, isoelectric point, and molecular weight, significantly influence the actual biological effects of molecules [12].

The outcomes of an in silico bioactivity analysis may diverge from in vitro results, necessitating a thorough exploration of diverse molecular parameters. Therefore, in databases such as BIOPEP and CAMPR_3_, which analyze a compiled literature of reported bioactive peptides, a comprehensive understanding of these molecular characteristics is essential for accurate predictions and interpretations of in silico findings in the context of experimental results.

#### 2.1.2. Computational Assessment of Peptide Aggregation Propensity

We utilized an in silico approach to definitively evaluate the aggregation propensities of peptides. The online software AGGRESCAN was utilized to calculate the number of “hot spots”, considering specific amino acids that could form aggregation-prone regions. Hydrophobic amino acids are commonly associated with inducing aggregation [12]. An in silico analysis of the peptides revealed zero hot spots, indicating low sequence aggregation propensity values (Table 1). This finding demonstrates that the peptides have a minimal tendency to form aggregates. However, Groot et al. [13] used β-amyloid peptide as a model to assess the individual aggregation tendencies of natural amino acids. Their research findings indicate the participation of hydrophobic amino acids, including isoleucine, phenylalanine, valine, and leucine, in the aggregation [13]. These amino acids are found in the sequences of peptides, specifically HdKNL (DPPNPKNL), HdVLPN (VYPFPGPLPN), and HdLPN (PFPGPLPN); however, they did not form aggregation-prone regions. Notably, HdVLPN and HdLPN exhibit low solubility (Table 2), which can be attributed to their composition being rich in hydrophobic amino acids.

In a molecular characterization, it was found that the HdKTLR peptide exhibited pronounced aggregation. The theoretical molecular weight of the peptide is 1588.4 Da; however, it increased to approximately 7836 Da under specific buffers with ionic strengths of 344 mM and 700 mM at pH 7.0 and pH 5.0, indicating a high degree of aggregation. An evaluation of the physicochemical parameters in Table 2 indicates that at pH 7.0, the peptide carries a net charge of −3. This charge may not be sufficiently high to cause repulsion between molecules, potentially leading to aggregation. Despite the HdKNL, HdLPN, and HdVLPN peptides having a net charge of 0 at pH 7.0, aggregation was still observed under conditions of high ionic strengths (344 mM and 700 mM) and at both pH levels of 7.0 and 5.0. This suggests that factors other than charge, such as ionic strength and hydrophobic interactions, may also significantly contribute to aggregation under these conditions.

Notably, peptide aggregation not only compromises their stability, resulting in a loss of activity, but also introduces other significant concerns, such as toxicity and immunogenicity [12]. However, several solutions exist to address this issue, such as designing analogs that incorporate a small number of amino acid substitutions, which could normalize or improve biological activity and reduce aggregation propensity [16].

### 2.2. In Vitro Validation of Peptide Bioactivities

#### 2.2.1. Antioxidant Activity

##### Antioxidant Assays: ORAC and ABTS

The antioxidant potential of *H. didactylus* peptides was assessed through oxygen radical absorbance capacity (ORAC) and 2,2′-Azinobis-(3-Ethylbenzthiazolin-6-Sulfonic Acid (ABTS) assays. The ORAC and ABTS assays assess distinct interactions between antioxidants and generated radicals. In the ORAC assay, antioxidants donate hydrogen to neutralize radicals, whereas in the ABTS assay, the reaction involves the transfer of electrons by the antioxidant [17]. The peptides HdVLPN and HdLPN exhibited antioxidant potential in both ABTS and ORAC assays (Table 3), establishing them as the two most promising candidates. In contrast, the HdKNL and HdPPP peptides demonstrated antioxidant activity exclusively in the ORAC assay, while the HdKTLR peptide did not show antioxidant potential in either the ABTS or ORAC assay. The HdVLPN and HdLPN peptides possess distinct amino acid compositions, where variations in the quantity and arrangement of hydrophobic amino acids (Val, Leu, and Pro for HdVLPN; Leu and Pro for HdLPN) and aromatic amino acids (Tyr and Phe for HdVLPN; Phe for HdLPN) may significantly influence their antioxidant potential [18].

Upon comparing our results with peptides previously identified in the ark shell (*Scapharca subcrenata*) by Jin et al. [19], it became evident that after converting their values to the same units as ours, our peptides exhibited superior antioxidant activity in both the ABTS and ORAC assays. Specifically, in the ABTS assay, P1 (MCLDSCLL) from the ark shell demonstrated an antioxidant activity of 0.678 µmol Trolox equivalents (TE)/µmol of peptide, and P2 (HPLDSLCL) exhibited an activity of 0.005 µmol TE/µmol of peptide. In the ORAC assay, the values were 0.334 µmol TE/µmol of peptide for P1 and 0.239 µmol TE/µmol of peptide for P2. Compared to the HdVLPN peptide, the ark shell peptides exhibited inferior antioxidant activity, achieving 1.560 µmol TE/µmol of peptide in the ORAC assay and 1.755 µmol TE/µmol of peptide in the ABTS assay. Additionally, the HdLPN peptide demonstrated antioxidant activity with 0.128 µmol TE/µmol in the ABTS assay, surpassing the activity of the P2 peptide.

In the food industry, the overproduction of free radicals, which leads to the generation of oxidants, can compromise food quality and shorten shelf-life [18]. Consequently, antioxidant peptides have proven to be a viable alternative to synthetic antioxidants as food preservatives, offering the advantage of mitigating the severe toxicity associated with synthetic antioxidants in food formulations [20]. Thus, the identified antioxidant bioactivity of *H. didactylus* peptides may have a potential application in this industry. Additionally, incorporating a balanced diet rich in antioxidants is believed to help mitigate the risk of chronic diseases associated with oxidative stress, such as Alzheimer’s disease, arthritis, heart disease, cancer, and others [18].

##### Influence of pH and Ionic Strength on Antioxidant Activity (ABTS Assay)

As revealed by the HPSEC analysis, the aggregation of peptides demonstrated sensitivity to varying ionic strength and pH conditions, which were factors further evaluated in relation to antioxidant activity using the ABTS assay. Our results demonstrated that HdKTLR exhibited no observable antioxidant activity. The antioxidant activity of the HdKNL peptide remained unaffected by a decrease in pH, whereas no antioxidant activity was observed in H_2_O, only in the presence of buffers (Figure 2A). However, for HdVLPN and HdLPN peptides, a reduction in pH resulted in a significant decrease in antioxidant activity (Figure 2B,C).

Interestingly, studies have unveiled that the reaction rates of specific amino acids—namely, Cys, Tyr, Trp, and His—in the ABTS scavenging assay exhibit a significant increase in ABTS radical scavenging values within a pH range of 5.0–7.0 [21]. These results reveal a strong pH dependence in the ABTS radical scavenging capacity of amino acids; namely, the values of Tyr increased from 2.89 µmol TE/µmol at pH 6.0 to 6.55 µmol TE/µmol at pH 8.0. In addition, this study suggests a minimum incubation time of 30 min for reliable Trolox equivalent values in the ABTS assay at pH 7.4 [21].

Our peptides, HdVLPN and HdLPN, demonstrated a significant correlation with pH variation, reflecting decreased ABTS radical scavenging values from pH 7.0 to 3.0. Specifically, at pH 7.0 and 700 mM, HdVLPN demonstrates values of 0.827 ± 0.021 µmol TE/µmol of peptide, whereas HdLPN shows 0.283 ± 0.177 µmol TE/µmol of peptide. Contrastingly, at pH 3 and 344 mM, HdVLPN records 0.163 ± 0.002 µmol TE/µmol of peptide, while HdLPN displays 0.012 ± 0.004 µmol TE/µmol of peptide (Figure 2B,C). The HdVLPN peptide includes the amino acid Tyr in its sequence, a characteristic that appears to align with the fact that the HdVLPN peptide and the amino acid Tyr exhibit greater antioxidant activity at pH 7.0. Interestingly, despite the peptide aggregates observed in the HPSEC chromatograms at pH 7.0, their antioxidant activity remained unaffected. On the contrary, they exhibited the greatest antioxidant activity, as demonstrated through the ABTS scavenging assay. However, additional studies are necessary to deepen our understanding of the behavior of peptides and their constituent amino acids at different pH levels. This is particularly important, as pH variation substantially impacts peptides’ antioxidant potential.

##### Mucus Interaction and Antioxidant Capacity (ORAC Assay)

We employed an experimental approach by mixing the peptides with the mucus to thoroughly investigate potential synergies. Therefore, we tested the combined effect of peptides and mucus on antioxidant activity using the ORAC assay (Figure 3 and Figure 4). This approach aimed to uncover any positive interactions, considering the diverse array of mucus molecules that could interact with the peptides.

Two mucus samples, obtained from wild and captive individuals, were used to evaluate the synergistic effect of the peptides. We pooled the collected mucus of several *H. didactylus* individuals, resulting in a protein value of 5888 ± 750 µg BSA/ mL for the captive sample and 111 ± 5 µg BSA/mL for the wild sample. Specifically, the HdVLPN peptide, when combined with mucus from captive individuals, exhibited a notable 31% increase in antioxidant activity (Figure 3D) compared with the sum of the bioactivity of the individual systems (mucus and peptide). The increment was even more pronounced at 44% for the wild mucus sample (Figure 4D). Likewise, HdLPN demonstrated significant enhancements, showing a substantial 51% increase in antioxidant activity when combined with mucus from captive individuals (Figure 3E) and a 54% increase for the wild mucus sample (Figure 4E). Furthermore, the HdKNL peptide displayed a remarkable 126% increase in antioxidant activity for the captive mucus sample (Figure 3B), and the HdPPP peptide showcased a 61% increase for the wild mucus sample (Figure 4C). The findings reveal a notable enhancement in antioxidant activity upon the incorporation of peptides into mucus.

The mucus comprises ca. 95% water and mucins, which are high molecular weight glycoproteins that are crucial in determining mucus viscosity [22,23]. Additionally, mucins encompass a diverse array of molecules with significant bioactivities. These include glycoproteins, glycosaminoglycans, pheromones, proteolytic enzymes, lectins, galectins, lysozymes, calmodulins, immunoglobulins, complements, C-reactive proteins, and antimicrobial peptides [24].

The diverse and abundant range of molecules within the mucus composition potentially contributes to interactions with peptides, namely, HdKNL, HdPPP, HdVLPN, and HdLPN, resulting in enhanced antioxidant activity through a synergistic effect. This synergistic combination of antioxidants could mirror a dynamic interplay, as highlighted in a study by Aklakur [25]. According to this study, pairing two antioxidants demonstrated a unique mechanism: one antioxidant reacted with the peroxy radical, being consumed in the process, while the second antioxidant could regenerate the first, preserving its functionality.

#### 2.2.2. Assessment of Antihypertensive Activity

The antihypertensive potential of bioactive peptides was tested through the inhibition of an angiotensin-converting enzyme (iACE) assay. The mucus of *H. didactylus* demonstrated an inhibitory effect, with an IC_50_ of 60 ± 7 µg of mucus protein/mL [4]. To identify the molecules responsible for this bioactivity, the peptides identified from the fraction were tested for their inhibitory properties. The iACE activity was only identified for the HdPPP peptide, which inhibited 50% of ACE at a concentration of 577 ± 90 µg of peptide/mL, representing the mean ± standard deviation (n = 3). This aligns with our previous article, which describes the peptides in in silico predictions using the BIOPEP database and identifies the HdPPP peptide as having the highest potential as an ACE inhibitor. The observed inhibitory potentials using BIOPEP were as follows: HdKTLR (A = 0.375), HdKNL (A = 0.125), HdVLPN (A = 1.200), HdLPN (A = 1.000), and HdPPP (A = 1.375). “A” represents the frequency of bioactive fragments in a protein or peptide [26].

When compared to other bioactive peptides derived from fish protein hydrolysates, such as those from *Syngnathus schlegeli*—specifically Thr-Phe-Pro-His-Gly-Pro (IC_50_ of 600 µg/mL) and His-Trp-Thr-Thr-Gln-Arg (IC_50_ of 1440 µg/mL)—and peptides from *Salmo salar*, including Ile-Trp, Ile-Tyr, Thr-Val-Tyr, Val-Trp, Val-Pro-Trp, and Val-Tyr, which exhibit IC_50_ values between 190 and 1040 µg/mL [27], the HdPPP peptide demonstrates comparable antihypertensive activity. Although the methodology differs, the HdPPP peptide also displays antihypertensive activity, aligning with the effects observed in these peptides from other fish sources.

The most widely used ACE blocker is the commercial drug Captopril^®^, which functions through the proline binding component to ACE at the following sites: Trp67, Asn68, Thr71, Asn72, Met340, and Arg348 [28]. The peptide HdPPP, rich in proline amino acids, emerges as our most promising candidate for exhibiting antihypertensive effects. Furthermore, the potential of this activity could be enhanced through gastrointestinal digestion. A study by Abachi et al. [29] provides examples of synthesized peptides where fragmentation and splitting during digestion resulted in di- and/or tri-peptides, enhancing their activity. For instance, the transformation of IKPLNY (IC_50_ 32 µg/mL) into IKP (IC_50_ 0.6 µg/mL) led to a remarkable 96% improvement in activity [28]. Likewise, for other peptides (HdKTLR, HdKNL, HdVLPN, and HdLPN) from which antihypertensive activity was undetermined, gastrointestinal digestion could have the potential to amplify their bioactivity via enzymatic hydrolysis. The increasing focus on peptides identified from natural sources is driven by concerns regarding common antihypertensive drugs’ adverse effects, including symptoms like coughing, taste disturbances, renal complications, and angioedema [27]. This shift towards natural peptides highlights the necessity for safer and potentially more effective alternatives in the treatment of hypertension.

#### 2.2.3. Evaluation of Antidiabetic Activity

The peptides were tested for antidiabetic activity based on the capacity to inhibit 50% of α-glucoside. The HdPPP was the only peptide that could inhibit 50% of α-glucoside activity at 1700 ± 40 µg of peptide/mL (mean ± standard deviation; n = 3). As predicted in silico by BIOPEP, HdPPP was the peptide with the highest probability of inhibiting α-glucoside (HdKNL A = 0.125, HdVLPN A = 0.100, and HdPPP A = 0.625). Although, compared to acarbose, which has an IC_50_ of 757 ± 75 µg/mL (mean ± standard deviation; n = 3), HdPPP proved less effective. Likewise, compared to other peptides hydrolyzed from food sources, namely, RVPSLM (23.05 µM) and TPSPR (40.02 µM) [29], the HdPPP peptide showed a higher IC_50_ of 2211 µM. As previously discussed, regarding antihypertensive activity, there is a potential for a similar augmentation of antidiabetic activity during digestion. The enzymatic hydrolysis during digestion results in the fractionation of peptides into di-peptides and tri-peptides, potentially generating active molecules that could inhibit α-glycoside.

#### 2.2.4. Antimicrobial Activity: Growth Inhibition Curves

In our prior investigation, the mucus collected from *H. didactylus* exhibited a notable inhibitory effect, with an approximately 76% inhibition against *Escherichia coli* ATCC 25922 at a concentration of 442 µg of mucus protein/mL and around 66% inhibition against *Listeria monocytogenes* NCTC 10357 at 221 µg of mucus protein/mL [4]. Considering the promising antimicrobial activity exhibited by the mucus of *H. didactylus*, growth inhibition curves were determined for strains of *E. coli* ATCC 25922, *Salmonella enterica* serovar Enteritidis ATCC 13076, *Pseudomonas aeruginosa* ATCC 27853, and *Staphylococcus aureus* ATCC 6538 using the peptides HdKTLR (262 µg/mL), HdKNL (262 µg/mL), HdPPP (130 µg/mL), HdVLPN (204 µg/mL), and HdLPN (227 µg/mL). The peptides did not demonstrate antimicrobial potential for the tested strains, except for the peptide HdPPP, which showed slight inhibition for the strain *S. aureus* (Figure 5). The HdPPP peptide at the tested concentration of 130 µg/mL has 5% DMSO, and to confirm that DMSO did not cause the inhibition of *S. aureus*, a control was made with only 5% DMSO, where no inhibition of the strain was observed.

In our previous study, we employed an in silico approach through the database CAMPR_3_ classifiers (SVM, RF, ANN, and DA) to predict the antimicrobial potential of peptides [4]. The HdKNL peptide notably surpassed the identified peptides, resulting in potential antimicrobial activity in all four classifiers. However, in vitro antimicrobial activity was not demonstrated for the HdKNL or the other peptides.

Generally, AMPs are cationic, with a net charge of +2 and up to +8 [30]. This allows the AMPs to interact with the negatively charged head groups of the microbial cytoplasmic membrane, adopting an amphipathic helical conformation that enables them to insert the hydrophobic face into the bilayer [31]. However, the peptides HdKNL, HdPPP, HdVLPN, and HdLPN have a net charge of 0 at pH 7, and the peptide HdKTLR has a net charge of −3 at pH 7, an atypical characteristic for AMPs, which aligns with their observed lack of antimicrobial activity. However, these peptides possess other characteristics that could contribute to antimicrobial activity, namely, the presence of proline in the sequence being particularly notable in some identified peptides. Studies have reported that proline hinges play a crucial role as a structural factor in facilitating the penetration of the molecule into the cell [32].

#### 2.2.5. Antibiofilm Activity Assessment

The peptides HdKTLR (262 µg/mL), HdKNL (262 µg/mL), HdPPP (130 µg/mL), HdVLPN (204 µg/mL), and HdLPN (227 µg/mL) were assessed for their efficacy in reducing biofilm formation after 24 h of incubation with strains of *E. coli*, *S. enterica*, *P. aeruginosa*, and *S. aureus* (Figure 6). When exposed to the *P. aeruginosa* strain, it was observed that the peptides exhibited significant differences compared to the control, resulting in substantially reduced biofilm production. However, this outcome did not occur with the other strains *(E. coli*, *S. enterica*, and *S. aureus*), as no significant differences were observed between the peptides and the control.

The reduction in biofilm production by *P. aeruginosa* through the action of peptides represents a promising outcome, particularly given the strain’s association with serious chronic lung diseases [33]. *P. aeruginosa* is particularly proficient at biofilm production [34]. Inhibiting biofilm formation during the mature phase offers substantial benefits, as microorganisms in biofilms exhibit heightened resistance to antimicrobial agents. Consequently, peptides could disrupt biofilm formation, transitioning bacteria from a biofilm state to a free-living one, thereby facilitating bacterial elimination [33]. Exploring additional approaches to address peptides’ relatively limited antimicrobial potential involves considering their combination with other antimicrobial agents. This strategy could enhance their efficacy, offering a promising pathway to strengthen antimicrobial activity.

#### 2.2.6. Evaluation of Peptides on Probiotic Growth and Viability

The effect of the peptides on the probiotics, particularly regarding their positive and negative interactions, was assessed for *Bifidobacterium* and *Lactobacillus* species. The *Bifidobacterium* tested included *B. longum* BG3, *B. animalis* Bo, *B. animalis* BB-12, *B. breve*, and *B. animalis* BLC. The *Lactobacillus* tested included *L. acidophilus* Ki, *L. plantarum* 299V, *L. casei*, *L. acidophilus* La5, and *L. rhamnosus* RII. The peptides (HdKTLR 262 µg/mL, HdKNL 262 µg/mL, HdVLPN 204 µg/mL, and HdLPN 227 µg/mL) did not inhibit the growth of probiotics and could instead promote growth (Figure 7 and Figure 8). The peptide HdPPP was not tested due to its solubility in DMSO, as this solvent is unsuitable for further applications in the food industry. Probiotics can benefit not only from the essential amino acids provided by the peptides, which could play a crucial role in modulating the integrity of bacterial cell membranes and walls [35].

Moreover, probiotics like *Lactobacillus lactis* depend on oligopeptides as a nitrogen source. Additionally, other interactions enhance the strain’s resistance to acidic conditions and promote probiotic growth by increasing protease or peptidase activity. An illustrative example is the Leu-Pro peptide, which boosts the protease activity of *Lactobacillus helveticus* [36]. Furthermore, the antioxidant properties of HdVLPN and HdLPN peptides can neutralize reactive oxygen species, which may help protecting probiotics from oxidative stress.

These results indicate a promising potential for applications in food products, such as acting as preservatives, since the peptides demonstrated no detrimental effects on the probiotics. Future research could focus on exploring the potential of bioactive peptides in modulating and preserving a healthy microbiome [36].

#### 2.2.7. Cytotoxicity Assessment on Caco-2 and HaCaT Cell Lines

Considering the prospective industrial applications of peptides, especially in the food industry, it is essential to assess the cytotoxicity of these peptides on the proliferation of human Caco-2 cells, which serve as a representative model for the intestine. Consequently, we conducted cytotoxicity tests on the two most promising peptides, HdVLPN and HdLPN, which were previously discussed regarding their significant antioxidant activity. These peptides were also subjected to a gastrointestinal simulation (see Section 2.2.8), which will also be necessary for investigating whether they are safe for humans to consume. The results demonstrate that both peptides (HdVLPN and HdLPN) exhibited no cytotoxic effects on human Caco-2 cells at tested concentrations from 25 to 100 μg/mL, with a metabolic inhibition below 30% (Figure 9A). This evaluation was crucial in determining the safety of these peptides for human consumption, particularly since they underwent a gastrointestinal simulation to ascertain potential bioaccessibility. Examining their impact on Caco-2 cells contributes to understanding their safety profile and provides valuable insights into their suitability for incorporation into human dietary applications.

To further investigate the potential benefits of the antioxidant peptides HdVLPN and HdLPN for skin health, we also utilized the HaCaT keratinocyte human cell line as an in vitro model to assess the cytotoxicity of the peptides. This cell line is extensively employed by researchers in skin biology studies as keratinocytes, constituting 95% of epidermal cells in the cutaneous layer [37,38], which play a fundamental role in skin function. The results demonstrate that both peptides (HdVLPN and HdLPN) exhibited no cytotoxic effects on human HaCaT cells at tested concentrations from 25 to 100 μg peptide/mL, with a metabolic inhibition below 30% (Figure 9B). These findings are promising, suggesting favorable applications for HdVLPN and HdLPN peptides as antioxidants for the skin.

Bioactive peptides derived from marine sources remain relatively underexplored for use in industrial products, such as functional foods, dietary supplements, medicines, and cosmetics [39]. Consequently, our finding holds promise for advancing the safe utilization of antioxidant peptides in these domains, offering a positive contribution towards their practical and beneficial implementation.

#### 2.2.8. Bioactivity Evaluation Post-Gastrointestinal Digestion: Antioxidant and Antihypertensive Effects

It was essential to assess the capacity of our bioactive peptides to resist the gastrointestinal tract and cross the intestinal epithelial barrier, thereby confirming their possible applicability as health potentiators [40]. The two most promising peptides, HdLPN and HdVLPN, exhibited significant antioxidant activity and were submitted to a gastrointestinal simulation at a concentration of 1.15 mg/mL for HdLPN and 1 mg/mL for HdVLPN. The digested content from the dialysis process, both the permeate and retentate, using 3.5 kDa of membranes, were evaluated in vitro for their antioxidant activity through the ORAC assay and only the permeate for antihypertensive potential through the inhibition of the angiotensin-I-converting enzyme (iACE).

The antioxidant activity of the two digested peptides, in both permeate and retentate forms, was comparable to the control (which utilized water in place of peptides); similar results were obtained for their antihypertensive activity in the permeate form. The results for both bioactivities show no significant differences when comparing the digested peptides’ retentate and permeate forms with the control (Figure 10). This suggests that the enzymatic hydrolysis occurring during digestion reduces the bioactivity of these peptides, as they exhibited antioxidant activity before digestion. Alternatively, the peptide concentration used may have been insufficient to observe the impact of digestion. Additionally, there was no subsequent activation or potentiation related to antihypertensive activity, which was also absent before digestion.

Therefore, a comprehensive analysis, including mass spectrometry, is essential to fully understand the impact of digestion on the hydrolysis of these peptides. Moreover, the exploration of biocompatible materials for safe delivery methods of peptides, e.g., encapsulation, to enable them to withstand gastrointestinal digestion and effectively reach target organs to exert their intended bioactivity should be conducted [40].

## 3. Materials and Methods

### 3.1. Materials and Reagents

The angiotensin-I-converting enzyme (peptidyl-di-peptidase A, EC 3.4.15.1, 5.1 U/mg); Trolox (6-hydroxy-2,5,7,8-tetramethyl-chroman-2-carboxylic acid); AAPH [2,2’-azo-bis-(2-methylpropionamidine)-dihydrochloride] chloramphenicol; pepsin, from porcine stomach mucosa (P7012, 500 U/mg); pancreatin, from porcine pancreas (P7545, 8 trypsin U/mg); bile salts (B863); and and DMSO (≥99.9%) were obtained from Sigma-Aldrich (St. Louis, MO, USA) and were used without further purification. α-glucosidase from *S. cerevisiae*, acarbose, and ρ-nitrophenyl-α-D-Glucopyranoside were obtained from Sigma-Aldrich (Sintra, Portugal). Fluorescein [3’,6’-dihydroxyspiro (isobenzofuran-1 [3H], 9’ [9H]-xanthen)-3-one] was purchased from Fisher Scientific (Hanover Park, IL, USA). The tri-peptide Abz-Gly-Phe (NO_2_)-Pro was obtained from Bachem Feinchemikalien (Bubendorf, Switzerland). Tris [tris (hydroxymethyl) aminomethane] was obtained from Honeywell Fluka (Charlotte, NC, USA). Mueller–Hinton (MH) and Man-Rogosa-Sharpe (MRS) broths were purchased from Biokar Diagnostics (Beauvais, France). The Pierce BCA Protein Assay Kit was purchased from Thermo Scientific (Vantaa, Finland). All remaining chemicals and reagents employed were of analytical grade or suitable for chromatography.

### 3.2. Sourcing and Preparing Peptides

The peptides HdKTLR (EDNSELGQETPTLR), HdKNL (DPPNPKNL), HdPPP (PAPPPPPP), HdVLPN (VYPFPGPLPN), and HdLPN (PFPGPLPN) were synthesized by GenScript, Piscataway, NJ, USA, with the purity ≥ 90%. The peptides were prepared following the guidelines provided in the technical datasheet, ensuring proper solubility and concentration, as detailed in Table 4.

### 3.3. Mucus Collection and Fish Source

The mucus of the body *H. didactylus* was collected according to the method described by [4]. Mucus was collected from *Halobatrachus didactylus* specimens caught by fishermen in central-west Portugal and were subsequently released by our team. Two pooled mucus samples were analyzed: one from five captive individuals at the Vasco da Gama Aquarium and the other from four wild individuals provided by the Portuguese Institute for Sea and Atmosphere (IPMA).

### 3.4. Soluble Protein Content

The Bicinchoninic Acid (BCA) methodology, using the Pierce BCA Protein Assay Kit, was employed to determine the soluble protein concentration in both the permeate and retentate forms of the samples obtained from digestion and dialysis. This methodology was also utilized to determine protein concentrations in the pooled mucus samples.

### 3.5. Antioxidant Activity

#### 3.5.1. ABTS Assay

ABTS radical activity is based on an antioxidant’s capacity to scavenge the ABTS’s oxidized state. For this, ABTS radical scavenging activity was performed in a 96-well microplate, following the method described by [4]. The absorbance was measured with a multidetection plate reader (Synergy H1, Winooski, VT, USA). For each sample, the standard and control analyses were duplicated. All measurements were performed in duplicate. The results are expressed in µmol of Trolox equivalents/µmol of peptide. To test the antioxidant activity under different pH and ionic strength conditions, the ABTS radical solution was prepared using the same buffers employed in the HPSEC analysis.

#### 3.5.2. ORAC Assay

The ORAC assay followed the protocol outlined by Coscueta et al. [41]. A black polystyrene 96-well microplate (Nunc, Roskilde, Denmark) was utilized, and measurements were taken using a multidetection plate reader (Synergy H1, Winooski, VT, USA) controlled by the Gen5 Biotek software version 3.04. Fluorescence was monitored for 80 min at 1 min intervals. Each sample, standard, blank, or control, analysis was duplicated. The final ORAC values were expressed as µmol TE/µmol of peptide for the peptides and µmol TE /mg of protein for the digestion retentate and permeate contents.

The synergistic interaction between the mucus and peptides was performed by mixing the peptides with mucus at a ratio of one part peptide to four parts mucus. In addition, control analyses were performed by mixing one part of the phosphate buffer with four parts of the mucus and another with one part of the peptide to four parts of the phosphate buffer. The mixtures were then tested in the ORAC assay. The final ORAC values were expressed as µM TE.

### 3.6. Antihypertensive Activity

The iACE assay was conducted following the procedure outlined by Coscueta et al. [41]. The assay was performed using a black polystyrene 96-well microplate (Nunc, Denmark), and fluorescence was monitored at 1 min intervals for 80 min using a multidetection plate reader (Synergy H1, Winooski, VT, USA) controlled by the Gen5 Biotek software version 3.04. Each sample, blank, and control, analysis was carried out in triplicate. The iACE was quantified as the concentration required to inhibit 50% of the enzymatic activity (IC_50_). The IC_50_ values were determined using non-linear modelling of the data obtained.

### 3.7. Antidiabetic Activity

The α-glucosidase inhibitory activity was evaluated through the following method, outlined by Vilas-Boas et al. [42]. The assay was performed in a 96-well microplate (Stansted) and incubated at 25 °C for 5 min, and the absorbance was monitored at 405 nm with a microplate reader (Synergy H1, BioTek Instruments, Winooski, VT, USA). A negative control was performed, with 50 μL of a buffer solution that replaced the peptides. Acarbose was used as a positive control at a 10 mg/mL concentration. Each sample and control analysis was triplicated. The inhibition of α-glucosidase was quantified as the concentration required to inhibit 50% of the enzymatic activity (IC_50_). The IC_50_ values were determined using non-linear modelling of the data obtained. The IC_50_ value for acarbose was determined to use as a comparative reference.

### 3.8. Growth Inhibition Curves

The antimicrobial activity of the mucus samples was evaluated following the method by [4] against the following pathogenic bacteria: Gram-negative *Escherichia coli* ATCC 25922, *Salmonella enterica* serovar Enteritidis ATCC 13076, and *Pseudomonas aeruginosa* ATCC 27853 and Gram-positive *Staphylococcus aureus* ATCC 6538. The mucus samples were combined with a 1/10 ratio of inoculum containing 1% (*v*/*v*) of bacteria cultured for 24 h in MH broth. For the positive control, deionized water was used instead of mucus samples with a 1% inoculum, while deionized water with MH served as the negative control. A control using chloramphenicol was made at a concentration of 16 µg/mL for *E. coli*, *S. enterica*, and *P. aeruginosa* and 100 µg/mL for *S. aureus*. The mixture was then transferred to a 96-well microplate (Sarstedt, Germany), and the optical density (OD) at 600 nm was measured every hour for a 24 h period at 37 °C using a microplate reader (Multiskan GO, Thermo Scientific, Finland). An increase in OD indicated bacterial growth.

### 3.9. Antibiofilm Activity

The antibiofilm activity was determined according to the following method, with some modifications [43]. Taking the microplate from 24 h inhibition growth curves, the contents of the plate were discarded, and each well was carefully washed three times with distilled water to remove non-adherent cells. The adhered cells were fixed with 150 µL of methanol for 10 min. Then, the methanol was discarded, the plates were left to dry, and then, the fixed biofilm was stained with 150 mL of crystal violet (CV) for 10 min. Excess stain was rinsed out by placing the plate under slow-running tap water. Once the plates were air-dried, the dye attached to the adherent cells was resolubilized by adding 150 µL of 33% (*v*/*v*) glacial acetic acid and agitated for 15 min. The optical density of the solution obtained was measured at 595 nm using a multidetection plate reader (Synergy H1, Winooski, VT, USA). All tests were carried out in duplicate, with a positive control using an inoculated culture in MH broth and a negative control using MH broth only. The results are presented as the percentage of biofilm formation, calculated according to the following formula, where the OD of the sample is the measured OD of the inoculated medium mixed with the peptides, and the OD of the positive control is the OD measured for the positive control.% Biofilm formation = (OD_sample_/OD_positive control_) × 100(1)

### 3.10. Probiotic Growth Curves

The effect of the peptides on probiotics was evaluated using the following probiotic *Lactobacillus* and *Bifidobacterium* species: *L. acidophilus* Ki (LAS), *L. acidophilus* La5, *L. casei* L26, *L. plantarum* 299V, *L. rhamnosus* RII, *B. longum* BG3 *B. animalis* spp. *lactis* BB12, *B. animalis* BLC, and *B. animalis* Bo. All the assays were performed using MRS broth supplemented with 0.5 g/L of L-cysteine [44]. The peptides were combined with a 1/10 ratio of inoculum, containing 2% (*v*/*v*) of the probiotic and cultured for 48 h in MRS broth. For the positive control, deionized water was used instead of the peptides, with a 2% inoculum, while deionized water with MRS served as the negative control. Peptide wells with *Bifidobacterium* were sealed with paraffin to assure anaerobiosis conditions. The bacterial growth was monitored at 650 nm for 24 h at 37 °C in a microplate reader (Multiskan GO, Thermo Scientific, Finland), and the optical density (OD) was measured hourly. Growth curves were created by transforming the absorbance values in natural logarithms ln (Abs).

### 3.11. Cytotoxicity Assay

To evaluate the cytotoxicity of the peptides HdVLPN and HdLPN, two cell lines were used, namely, caucasian colon adenocarcinoma cells—Caco-2 (ECACC 86010202) and 5—and human keratinocytes—HaCaT (300493, CLS, Eppelheim, Germany). The assay was carried out as described by [45]. Afterward, the culture media were carefully removed and replaced with 25, 50, and 100 μg/mL of peptide (sterile filtered). The cytotoxicity was evaluated using the PrestoBlue HS Cell Viability assay (Thermo Scientific, Waltham, MA, USA), following the protocol described by the manufacturer. Fluorescence was measured using a fluorescence excitation wavelength of 560 nm and an emission of 590 nm by the multidetection plate reader Synergy H1 (BioTek Instruments, Winooski, VT, USA) controlled by the Gen5 BioTek software version 3.04. The metabolic inhibition was determined as previously described by Coscueta et al. [46]

### 3.12. Gastrointestinal Simulation

The gastrointestinal simulation was based on the standardized static digestion model INFOGEST 2.0 protocol [46]. The procedure is divided into three phases, the oral, gastric, and intestinal, with the corresponding fluids mimicking the in vivo conditions. Therefore, the oral phase consisted in the dilution of the peptides HdVLPN (1 mg/mL) and HdLPN (1.15 mg/mL) at 1:1 (*v*/*v*) with simulated salivary fluid and agitated in the orbital shaker MaxQ 6000 at 200 rpm and at 37 °C for 2 min for mastication simulation. After this, the oral phase was diluted at 1:1 (*v*/*v*) with simulated gastric fluid and gastric enzyme, namely, pepsin that was prepared according to INFOGEST protocol and agitated at 130 rpm and at 37 °C, at pH 3.0 for 2 h. The resultant gastric phase was subsequently diluted at 1:1 (*v*/*v*) with simulated intestinal fluid, bile salts, and pancreatic enzymes (pancreatin from the porcine pancreas) and incubated at pH 7.0 for 2 h, at 45 rpm and at 37 °C. For each sample two replicas were performed, and a negative control without a sample, using ultrapure water, was used.

After digestion, to assess the possible absorption of the peptides, a dialysis process was performed using 3.5 kDa of membranes (cellulose membranes in dialysis tubes). Therefore, the final content of the digestion was poured inside the membrane (8 mL of the sample), and a significant volume of water was added to the outside, covering the sample inside the membrane (70 mL of ultrapure water). The dialysis samples were placed in an orbital shaker at 50 rpm and at 37 °C overnight. After the incubation time, the contents of the membrane (retentate) and the water (permeate) were collected and lyophilized. The lyophilized contents from the retentate were reconstituted in 4 mL of ultrapure water and from the permeate were reconstituted in 8 mL of ultrapure water.

### 3.13. Molecular Size Validation

The peptides’ molecular size validation was determined using high-performance size exclusion chromatography (HPSEC). An Agilent AdvanceBio SEC column (Agilent Technologies, London, UK), 2.7 µm in particle size, 130 Å in pore size, and 7.8 in inner diameter × 300 mm in length, was used. The column was eluted isocratically using various buffers with different pH and ionic strengths, namely, 344 mM at pH 7 (0.15 M of a phosphate buffer with NaH_2_PO_4_), 700 mM at pH 7 (0.15 M of a phosphate buffer with NaH_2_PO_4_ and NaCl), 344 mM at pH 5 (0. 15 M of an acetate buffer with acetic acid and NaCl), 700 mM at pH 5 (0.15 M of an acetate buffer with acetic acid and NaCl), 344 mM at pH 3 (0.05 M of a formate buffer with fomic acid and NaCl), and 700 mM at pH 3 (0.05 M of a formate buffer with formic acid and NaCl) at a flow rate of 1 mL/min. The sample injection volume was 10 µL. The instrument used was Waters 2690 with a photodiode array detector (PDA 190–600 nm). The software Empower 3 was used for data collection. To determine the molecular weights of the resulting chromatogram peaks, a calibration curve was made with the following protein standards: Ovalbumin (44,300 Da), Myoglobin (17,600 Da), Cytochrome C (12,327 Da), Aprotinin (6511 Da), Neurotensin (1672 Da), Angiotensin-II (1040 Da), Tyr-Phe dipeptide (328.4 Da), and L-tryptophan (204 Da).

### 3.14. Statistical Analysis

Different tests were applied for analyzing the data. Kruskal–Wallis was used for the ORAC assay, Mann–Whitney U for the ABTS assay, and repeated measures with Bonferroni post hoc for antimicrobial growth curves. For the antibiofilm assay, a one-way ANOVA was applied for *E. coli* and *P. aeruginosa* and Kruskal–Wallis for *S. aureus* and *S. enterica*. Cytotoxicity was assessed with a one-way ANOVA, followed by Tukey’s test. The antioxidant activity of the digested peptides was evaluated using the Kruskal–Wallis test for the retentate form and a one-way ANOVA, followed by Tukey’s test, for the permeate form. The iACE activity of the digested peptides was evaluated using the Kruskal–Wallis test for the permeate form. The effects of pH and ionic strength on ABTS were analyzed using Kruskal–Wallis for the peptides HdVLPN and HdLPN and a one-way ANOVA with Tukey’s test for the peptide HdKNL. Lastly, the mucus–peptide interaction was analyzed with *t*-tests. All analyses were performed using SPSS 28.0.0.0 (SPSS Inc., Chicago, IL, USA).

## 4. Conclusions

Our study delves into using in silico methodologies to predict peptide bioactivities, followed by their validation through in vitro assays. Employing in silico techniques for the bioactivity prediction of novel peptides offers a promising and efficient path for a preliminary exploration, although validation through in vitro assays remains imperative.

This study identifies five bioactive peptides through in silico prediction: HdPPP with the PAPPPPPP sequence, HdVLPN with the VYPFPGPLPN sequence, and HdLPN with the PFPGPLPN sequence. In vitro studies have revealed that HdPPP exhibits antihypertensive and antidiabetic properties, while HdLPN and HdVLPN demonstrate significant antioxidant activity. However, the bioactivities of these peptides could face limitations and not reach their full potential due to their tendency to aggregate under the physiological conditions of pH 7 with ionic strengths of 700 mM and 344 mM.

Consequently, a comprehensive study was conducted to uncover this issue, discovering a valuable interaction between the peptides and mucus. It appears that the molecules present in mucus can augment the bioactivity of peptides through synergistic interactions, thereby optimizing their conformation for enhanced functionality and/or in neutralization oxidants by facilitating electron and proton donation to stabilize the peptides in antioxidant activity.

Further investigations are imperative to delineate the optimal conditions for peptide bioactivity. Subsequent research should explore encapsulation techniques and the utilization of matrices mimicking mucins to enhance peptide bioactivity.

Overall, this study highlights the potential of the marine environment, and in particular marine fish, to contribute to the bioprospection of novel bioactive molecules with wide applications in food, pharmaceutical, nutraceutical, and cosmeceutical products.

## 5. Patents

New peptides and uses thereof: Instituto Nacional da Propriedade Industrial (patent Nº.: PT118365; 5 December 2022).

## Figures and Tables

**Figure 1 marinedrugs-23-00082-f001:**
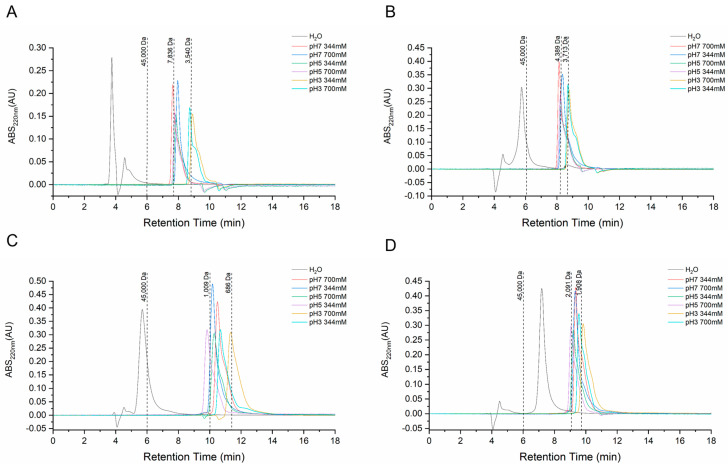
SEC chromatograms of the peptides at different ionic strength and pH conditions: HdKTLR (**A**), HdKNL (**B**), HdVLPN (**C**), and HdLPN (**D**).

**Figure 2 marinedrugs-23-00082-f002:**
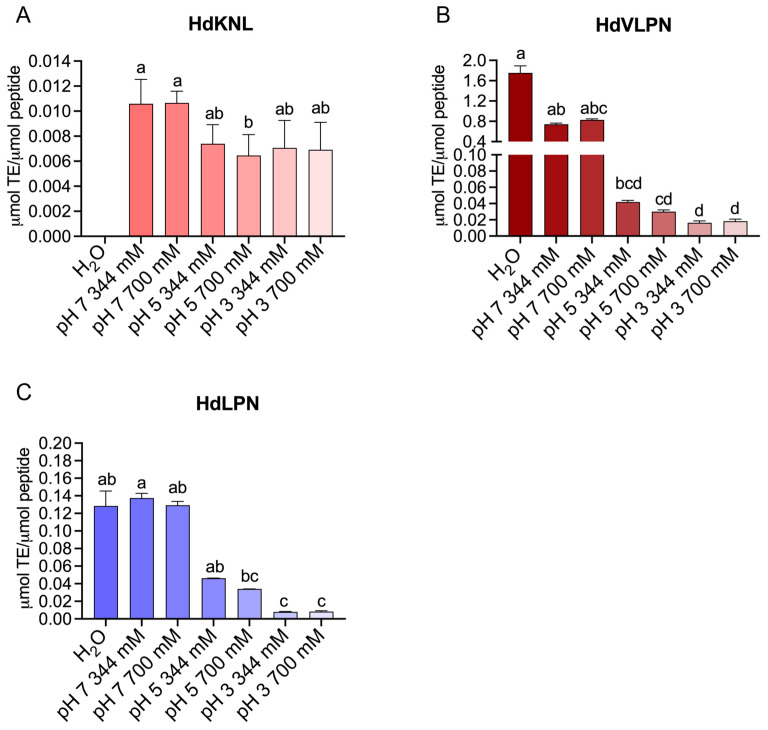
Evaluation of antioxidant activity through the ABTS scavenging assay at different ionic strengths and pH conditions of the peptides: HdKNL (**A**) (no bar is shown for H_2_O, as no antioxidant activity was detected); HdVLPN (**B**); and HdLPN (**C**). All values are means (n ≥ 3) ± standard deviation (SD). Different lower-case letters (a–d) at each condition indicate that the values differed significantly (*p* < 0.05). Data (**B**,**C**) were analyzed using the Kruskal–Wallis test and (**A**) a one-way ANOVA with Tukey’s test.

**Figure 3 marinedrugs-23-00082-f003:**
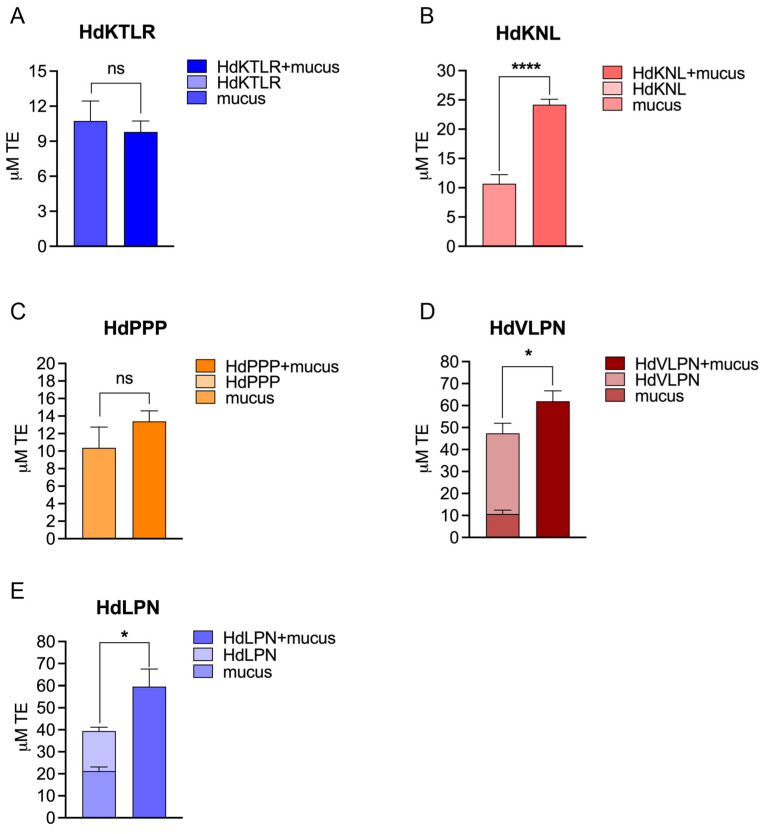
Evaluation of the synergistic effect between peptides and the captive mucus sample through the ORAC assay. HdKTLR (**A**), HdKNL (**B**), HdPPP (**C**), HdVLPN (**D**), and HdLPN (**E**). In graphs (**A**–**C**), no bar is shown for the peptides, as no antioxidant activity was detected. All values are means (n ≥ 3) ± standard deviation. ns: *p* > 0.05, * *p* < 0.05, and **** *p* < 0.0001. Data were analyzed using a *t*-test.

**Figure 4 marinedrugs-23-00082-f004:**
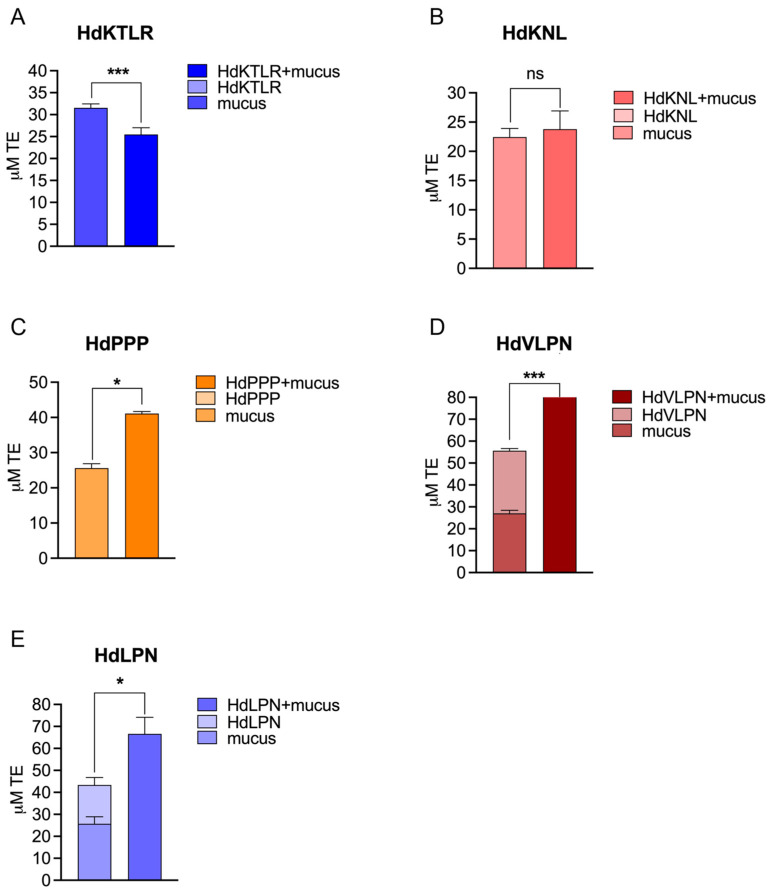
Evaluation of the synergistic effect between peptides and the wild mucus sample through the ORAC assay. HdKTLR (**A**)**,** HdKNL (**B**), HdPPP (**C**), HdVLPN (**D**), and HdLPN (**E**). In graphs (**A**–**C**), no bar is shown for the peptides, as no antioxidant activity was detected. All values are means (n ≥ 3) ± SD. ns: *p* > 0.05, * *p* < 0.05, and *** *p* < 0.001. Data were analyzed using a *t*-test.

**Figure 5 marinedrugs-23-00082-f005:**
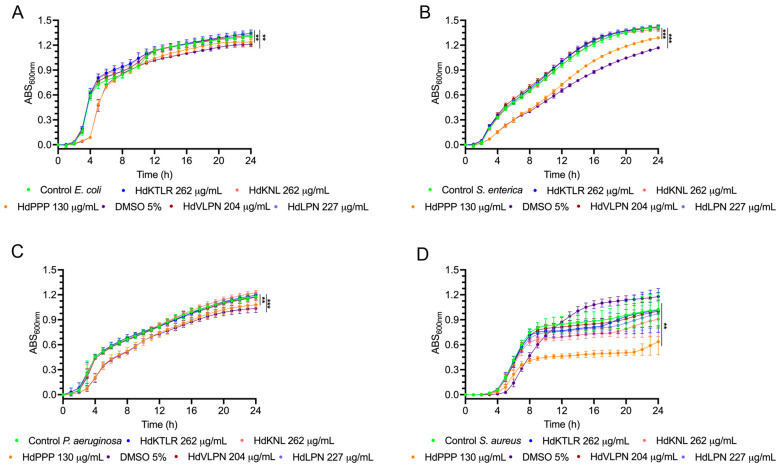
Inhibition growth curves of peptides HdKTLR, HdKNL, HdVLPN, and HdPPP against pathogenic bacteria: *E. coli* (**A**), *S. enterica* (**B**), *P. aeruginosa* (**C**), and *S. aureus* (**D**). ** *p* < 0.01, *** *p* < 0.001. Data were analyzed using repeated measures with a Bonferroni post hoc test.

**Figure 6 marinedrugs-23-00082-f006:**
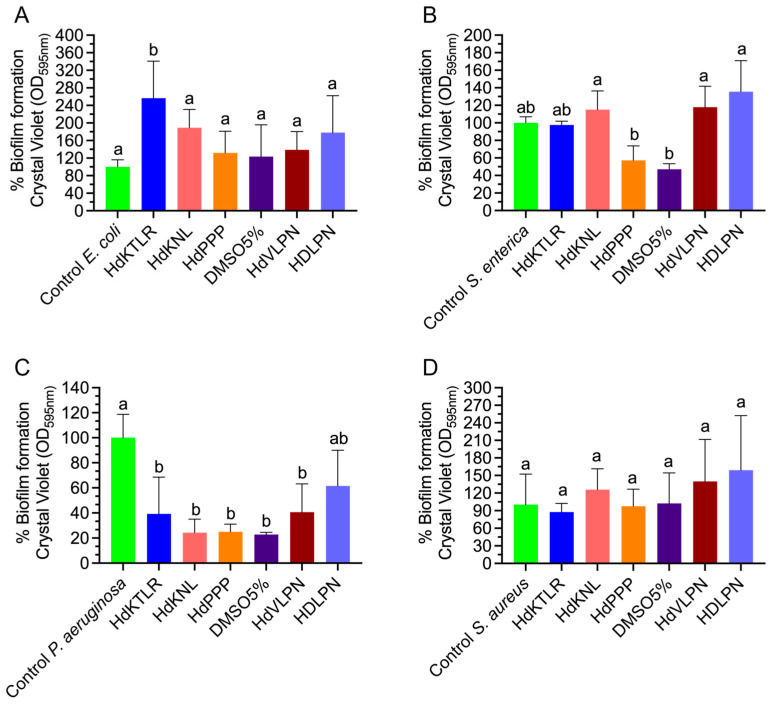
Percentage of biofilm formation of the strains *E. coli* (**A**), *S. enterica* (**B**), *P. aeruginosa* (**C**), and *S. aureus* (**D**) in the presence of peptides. All values are means (n ≥ 3) ± standard deviation (SD). Different lower-case letters (a,b) at each condition indicate that the values differed significantly (*p* < 0.05). Data were analyzed using one-way ANOVA for *E. coli* and *P. aeruginosa* and Kruskal–Wallis for *S. aureus* and *S. enterica*.

**Figure 7 marinedrugs-23-00082-f007:**
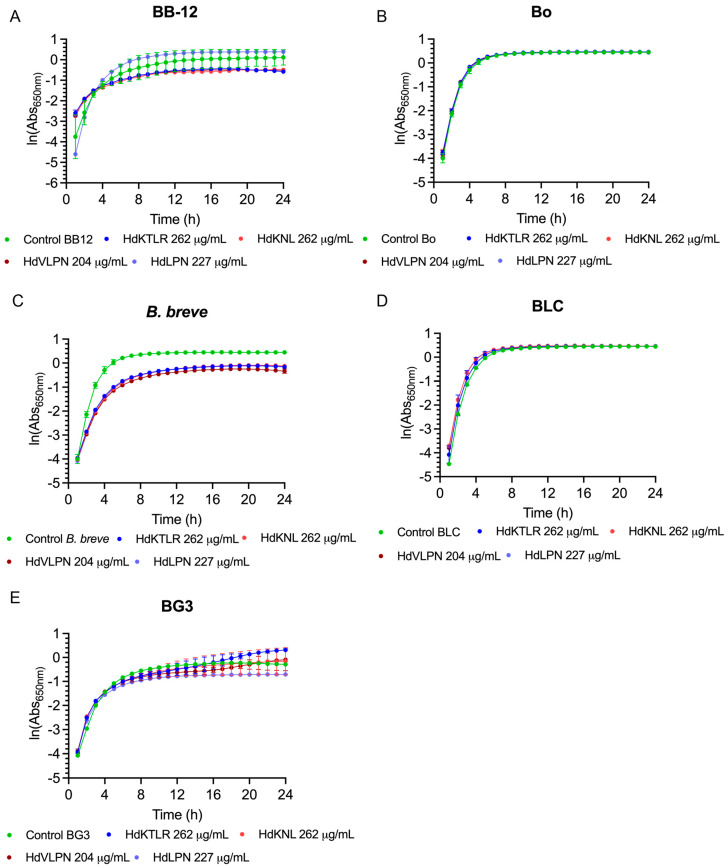
Growth curves of peptides HdKTLR, HdKNL, HdVLPN, and HdLPN with the probiotics of *Bifidobacterium* species: *B. animalis* BB-12 (**A**), *B. animalis* Bo (**B**), *B. breve* (**C**), *B. animalis* BLC (**D**), and *B. longum* BG3 (**E**).

**Figure 8 marinedrugs-23-00082-f008:**
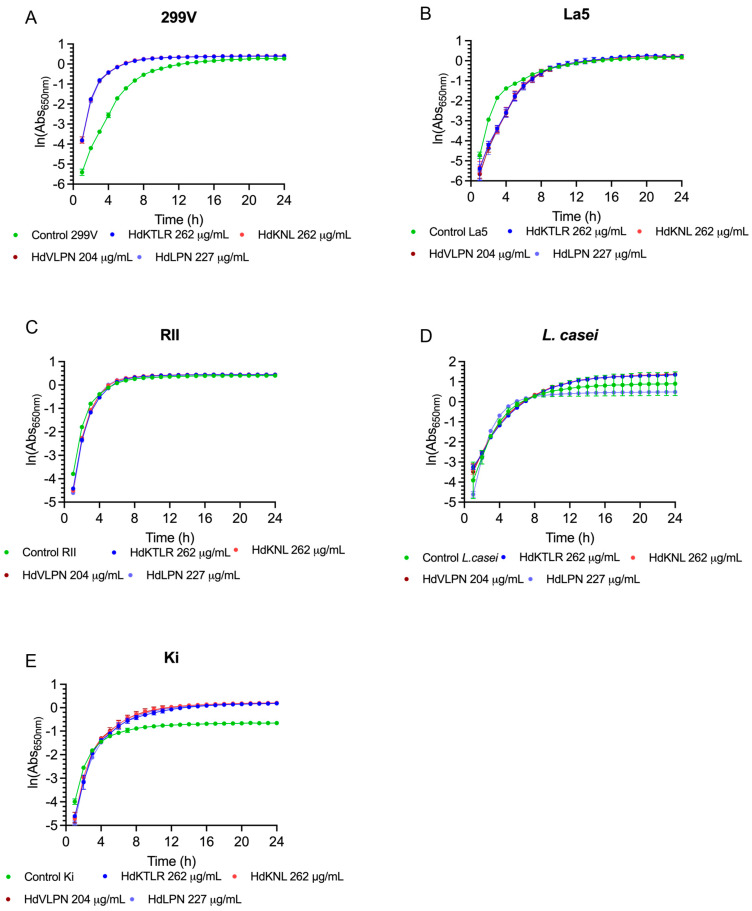
Growth curves of peptides HdKTLR, HdKNL, HdVLPN, and HdLPN with the probiotics of *Lactobacillus* species: *L. plantarum* 299V (**A**), *L. acidophilus* La5 (**B**), *L. rhamnosus* RII (**C**), *L. casei* (**D**), and *L. acidophilus* Ki (**E**).

**Figure 9 marinedrugs-23-00082-f009:**
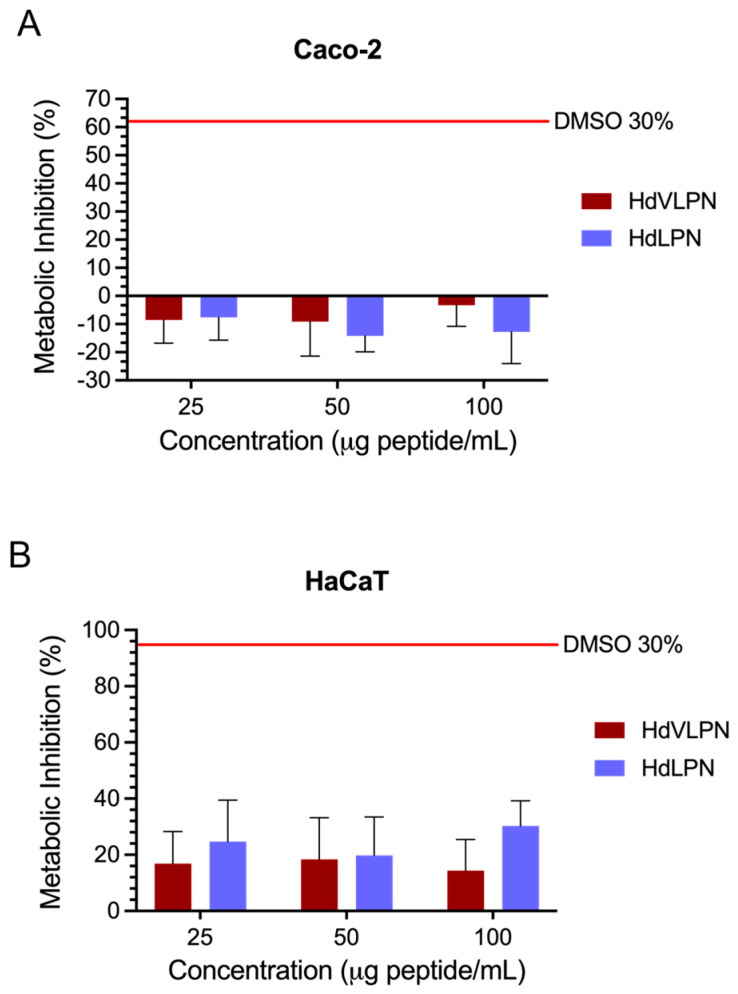
Cytotoxicity evaluation of HdVLPN and HdLPN peptides of human Caco-2 cell line (**A**) and HaCaT human keratinocyte cell line (**B**). All values are means (n ≥ 3) ± standard deviation (SD). There was no significant difference (*p* > 0.05) in metabolic inhibition between the two peptides at the range of concentrations tested (25 to 100 μg of peptide/mL) for Caco-2 and HaCaT cells, though these differed significantly from DMSO 30% (*p* < 0.001). Data were analyzed using one-way ANOVA, followed by Tukey’s test.

**Figure 10 marinedrugs-23-00082-f010:**
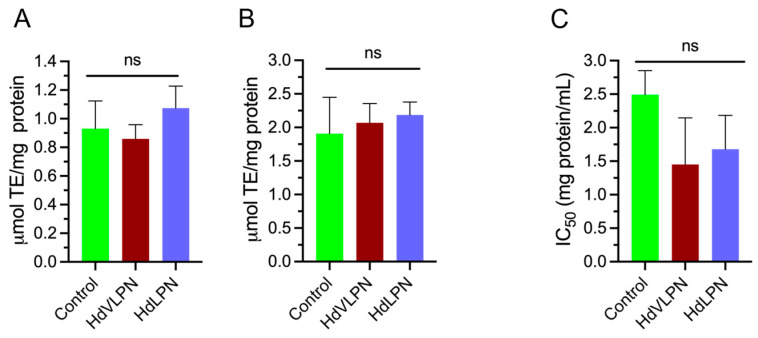
Antioxidant and inhibitory activities during the final phase of peptide digestion and dialysis: ORAC antioxidant activity in the retentate (**A**) and permeate forms (**B**). No significant differences were observed between peptides and the control in both the retentate and permeate forms (*p* > 0.05). Data for the retentate were analyzed using the Kruskal–Wallis test, while a one-way ANOVA, followed by Tukey’s post hoc test was applied to the permeate. An iACE 50% inhibition concentration from dialysis permeates during the final digestion phase (**C**). Similarly, no significant differences were observed between peptides and the control (*p* > 0.05), with data analyzed using the Kruskal–Wallis test.

**Table 1 marinedrugs-23-00082-t001:** Determination of aggregation propensities by AGGRESCAN [14].

Peptides	Sequence Average Aggregation Propensity Value (a^3^vSA)	Hot Spot Number
HdKTLR	−0.612	0
HdKNL	−0.624	0
HdPPP	−0.297	0
HdVLPN	0.271	0
HdLPN	−0.0005	0

**Table 2 marinedrugs-23-00082-t002:** Physicochemical parameters determined by INNOVAGEN [15].

Peptides	Extinction Coefficient M^−1^cm^−1^	Isoelectric Point	Net Charge at pH 7	Estimated Solubility
HdKTLR	0	pH 3.54	−3	+
HdKNL	0	pH 6.66	0	+
HdPPP	0	pH 4.26	0	−
HdVLPN	1280	pH 3.63	0	−
HdLPN	0	pH 4.16	0	−

+: good water solubility; −: poor water solubility. Water solubility was estimated according to the isoelectric point, the number of charged residues, and the peptide length.

**Table 3 marinedrugs-23-00082-t003:** Bioactivities: ORAC and ABTS.

Peptides	Sequence	Molecular Weight (Da)	ORAC (µmol TE/µmol Peptide)	ABTS (µmol TE/µmol Peptide)
HdKTLR	EDNSELGQETPTLR	1588.4	nd	nd
HdKNL	DPPNPKNL	893.8	0.048 ± 0.002 ^a^	nd
HdPPP	PAPPPPPP	768.8	0.054 ± 0.006 ^a,b^	nd
HdVLPN	VYPFPGPLPN	1099.6	1.560 ± 0.110 ^c^	1.755 ± 0.138 ^a^
HdLPN	PFPGPLPN	837.5	0.195 ± 0.030 ^b,c^	0.128 ± 0.017 ^b^

nd: not detected. All values are means (n ≥ 3) ± SD. ^a,b,c^ The different letter superscripts in the same column indicate that the values differed significantly (*p* < 0.05). ORAC data were analyzed with the Kruskal–Wallis test and ABTS data with the Mann–Whitney U test.

**Table 4 marinedrugs-23-00082-t004:** Summary of peptide solubility and concentration values.

Peptides	Sequences	Molecular Weight (Da)	Concentration (mg/mL)	Solubility
HdKTLR	EDNSELGQETPTLR	1588.4	2.6	H_2_O
HdKNL	DPPNPKNL	893.8	2.6	H_2_O
HdPPP	PAPPPPPP	768.8	2.6	DMSO
HdVLPN	VYPFPGPLPN	1099.6	2.0	H_2_O
HdLPN	PFPGPLPN	837.5	2.3	H_2_O

## Data Availability

The data presented in this study are available on request from the corresponding author. The data are not publicly available, as the patenting process is still ongoing at the time this article was submitted.

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
