# Peer review of "Bioprospecting Bioactive Peptides in Halobatrachus didactylus Body Mucus: From In Silico Insights to Essential In Vitro Validation"

_marinedrugs, 2025, doi:10.3390/md23020082_

Round 1

Reviewer 1 Report

Comments and Suggestions for Authors

Author Response

There are several points to be addressed by the authors.

  1. Abstract: please check the abbreviated names of two types "HdLETTERS" and"HDLETTERS"; it seems that a lower-case letter "d" is proper being used throughout the main text of the manuscript.

R: We thank the reviewer for highlighting this aspect in the abstract, we have changed the “HDLETTERS” to the lower-case “d” (highlithed in red).Line 16. The “Hd” comes from the Halobatrachus didactylus.

  1. Results and Discussion: "Peptides prone to aggregation typically show broader peaks and larger peak areas due to the formation of larger complexes" (Line 111): It is not so adequate to describe aggregated peptides as "complexes" whose strict chemical terminology is accepted. Besides, broader chromatographic peaks are naturally of larger areas, that, by the way, does not require additional repetitions, as at Lines 113-114. Rather, broader peaks might prove excessive interaction of the analyte (peptide) with the stationary phase followed by a stronger retention in the column. Upper threshold for acceptable accuracy of HPLC-like technique has to be in mind for every analyte parameter.

R: We appreciate your comment and would like to adress the redundancy of our discussion. Therefore, we have revised the discussion by addressing the impact of the mobile phase in aggregation with literature review (highlited in red).Line 112-121.

  1. Table 2: a solvent in which a solubility value is determined, should be pointed out. Thus, the "estimated solubility in water" (if applicable) has to be in the Table 2. Besides, the notes should report what is for "+" and "-" in this Table, and what are detection limits for solubility value? The peptide HdPPP is soluble exclusively in pure DMSO (Line 87), and what about HdVLPN and HdLPN, which occurred with a sign "-"?

R: Thank you for your observation regarding this gap in Table 2. We would like to clarify that the estimated solubility by INNOVAGEN refers to solubility in water. In our table, the “+” indicates 'Good water solubility,' while the “-” represents 'Poor water solubility,' as stated by INNOVAGEN. It is also stated that water solubility is estimated based on the isoelectric point, the number of charged residues, and the peptide length. To ensure clarity, we will highlight the meanings of “+” and “-” in the table footer highlighted in red. Line 177-178.

  1. As a reader could understand, the online software AGGRESCAN and INNOVAGEN failed to adequately assess the aggregation propensity and solubility, respectively, of the peptides under study. What recommendations from the authors these outcomes are followed by? Might those be an avoidance of using this software in respect to the definite group of peptides, thus with a specific set of structural/chemical characters?

R: Thank you for your comment regarding this issue and the relevant questions. The AGGRESCAN server calculates the a3vSA based on the a3v values determined for each amino acid. The software then uses these values to assess the aggregation tendency of a given peptide sequence. To achieve this, the tool relies on a database of 57 amyloidogenic proteins, where aggregation hot spots have been experimentally identified. This data was used to train the model and validate its predictions (Conchillo-Solé, O., de Groot, N.S., Avilés, F.X. et al. AGGRESCAN: a server for the prediction and evaluation of "hot spots" of aggregation in polypeptides. BMC Bioinformatics 8, 65 (2007). https://doi.org/10.1186/1471-2105-8-65). The INNOVAGEN peptide property calculator highlights the algorithm's limitations in calculating net charge based on pH (http://pepcalc.com). These softwares have some limitations and the collection of aditional data could improve the accuracy of training models.

In our opinion, the AGGRESCAN model is highly useful for preliminary analyses and validation of protein and peptide aggregation. In fact, could be a valuable resource in studies on diseases associated with proteins and peptide aggregation, such as Alzheimer's disease, Parkinson's disease, and type II diabetes (Tu, Maolin, et al. "Advancement and prospects of bioinformatics analysis for studying bioactive peptides from food-derived protein: Sequence, structure, and functions." TrAC Trends in Analytical Chemistry 105 (2018): 7-17). Therefore, it may also be useful for predicting the aggregation propensity of bioactive peptides, as their bioactivity can be affected by aggregation (Zapadka, Karolina L., et al. "Factors affecting the physical stability (aggregation) of peptide therapeutics." Interface focus7.6 (2017): 20170030).

The in silico tests demonstrated no propensity for aggregation of our peptides, whereas in vitro analysis showed the formation of aggregates. We recommend conducting in vitro tests to validate aggregation and evaluate whether it affects bioactivity, considering the origin of the peptides. The peptides in question were identified from the mucus of H. didactylus, which interacts with its surrounding environment, including specific pH conditions and ionic interactions with sea water, and interactions with the molecules present in the mucus. In our article, we demonstrated that there is a synergy between the peptides and mucus that enhances antioxidant activity. Thus, assessing aggregation in peptides should not only involve evaluating the amino acid sequence for aggregation propensity but also consider their interactions with the environment.

  1. Discussion at Lines 163-177 contains some confusing points: beginning with measures vs peptide aggregation, that causes negative consequences, it is followed by "noteworthy solution" of transforming the peptide aggregates "into nanoparticles for the bioactive delivery of curcumin". Firstly, please evaluate sizes of nanoparticles and peptides di-, tri-, or tetramers proved for HdKTLR, HdKNL, HdVLPN, HdLPN. Secondly, this oppositely directed movement toward aggregation to occur peptidic nanostructures is not related to all the bioactivities discussed below this text portion, since the bioactivities are exhibited by non-aggregated peptides, at least those under study within the manuscript. The authors wrote themselves that bioactivities of these peptides could face limitations and not reach their full potential due to their tendency to aggregate (Line 718). Please concern a necessity of such distractions.

R: We appreciate your comment regarding the discussion in Lines 163-177. Thus, and we decided to remove this section to avoid confusion regarding the aim of our study. The transformation of peptides into nanoparticles is not the focus of our research; our objective is to assess the bioactivity of the peptides in vitro, particularly concerning the effects of aggregation. Therefore, the formation of peptides into nanostructures is not relevant in this context.

  1. Conclusion: "using in silico methodologies to discover peptide bioactivities" seems hardly deserving of a serious position within this section, if any position there. Are the bioactivities under question (antioxidant, pharmaceutical, etc.) discovered actually by using in silico techniques; I'm afraid not at all.

R: Thank you for your comment. We acknowledge your concern and would like to clarify that the discovery of new bioactive peptides is achieved through a combination of in silico and in vitro studies. Therefore, we will revise the phrase “using in silico methodologies to discover peptide bioactivities” to “using in silico methodologies to predict peptide bioactivities”, “predict” highlited in red (Line 712) and other modifications highlited in red Line 714. This adjustment better reflects our approach, as in silico analysis helps identify potential candidates, which are then validated experimentally. For instance, the HdPPP peptide was predicted to have enzymatic inhibition activity through in silico analysis and was subsequently confirmed in vitro.

Other proposed corrections to manuscript

  1. Line 60: "ACE" should be spelled out as the abbreviation at its first entry in the main text, namely "angiotensin-converting enzyme (ACE)".

R:  Thank for your suggestion. We changed it in the article highlited in red (Line 59).

  1. Table 2: "Iso–electric point" should be replaced by "Isoelectric point"

R:  Thank for your suggestion. We chnaged it in the article highlited in red.

Reviewer 2 Report

Comments and Suggestions for Authors

Comments

The article discusses the protein nature of Halobatrachus didactylus mucus and the relationship between the bioactivity of peptides and their characteristics.

There are some comments on the text and content of the article.

1. The introduction to the article is very brief and does not cover the problem of studies of marine life mucus conducted earlier using other representatives of the marine fauna as examples. The authors in the introduction refer only to their own previously conducted studies (reference [4]). In fact, other marine life secretes similar mucus: fish, squid, hagfish, etc. In the discussion, the authors refer to these works, for example, for a number of fish (references [8, 24,26, 35] for hagfish [10, 34].

In my opinion, the introduction should discuss in detail the previously conducted and described in the literature studies on the proteins of marine life mucus.

2. The authors present the results of the analysis of the molecular weight of peptides by the HPSEC method in the form of GPC curves, and on these curves they indicate the values ​​of the molecular weights corresponding to the peaks on the GPC curve.

Question: why not show the MWD curves in the figures? This is much clearer.

3. Based on the data from the analysis of the molecular weight of peptides by the HPSEC method, the polydispersity coefficients of the polymer MW are usually calculated. This is interesting information and is usually discussed in the text of the article.

Why are the polydispersity coefficients not presented and discussed in the article peptides?

4. The captions of figures 2, 3, 5, 9, 10 contain very long texts. Some of them can be transferred to the text of the article.

5. The degree of purity of dimethyl sulfoxide should be indicated.

Author Response

There are some comments on the text and content of the article.

  1. The introduction to the article is very brief and does not cover the problem of studies of marine life mucus conducted earlier using other representatives of the marine fauna as examples. The authors in the introduction refer only to their own previously conducted studies (reference [4]). In fact, other marine life secretes similar mucus: fish, squid, hagfish, etc. In the discussion, the authors refer to these works, for example, for a number of fish (references [8, 24,26, 35] for hagfish [10, 34]. In my opinion, the introduction should discuss in detail the previously conducted and described in the literature studies on the proteins of marine life mucus.

R: Thank you for your comment. We would like to clarify that this article is a continuation of our previous study (reference [4]: Fernandez Cunha, Marta, et al. “Exploring bioactivities and peptide content of body mucus from the Lusitanian toadfish Halobatrachus didactylus.” Molecules 28.18 (2023): 6458), where we provided an overview of the role of mucus in other fish species within the introduction. The objective of this article is to validate the bioactivities of peptides previously identified in the mucus of H. didactylus, whose bioactivities were predicted in silico. Given the continuity of our research, we chose to maintain a focused introduction, referencing our prior work where the broader context of marine mucus studies was already discussed. Therefore, we believe that the current introduction adequately serves its purpose within the scope of this study.

  1. The authors present the results of the analysis of the molecular weight of peptides by the HPSEC method in the form of GPC curves, and on these curves they indicate the values of the molecular weights corresponding to the peaks on the GPC curve. Question: why not show the MWD curves in the figures? This is much clearer.

R: Thank you for your comment. It is not clear whether the reference to MWD curves refers to the chromatograms of the standards used or to something else. Likewise, we clarify that this type of graph is a typology that we are already accustomed to use and that has been efficient in transmitting the information, as you can see in the articles: https://doi.org/10.3390/molecules28186458 ; https://doi.org/10.1016/j.algal.2024.103695 ; https://doi.org/10.1016/j.foodres.2022.111549 ; https://doi.org/10.1002/biot.202300291 ;

  1. Based on the data from the analysis of the molecular weight of peptides by the HPSEC method, the polydispersity coefficients of the polymer MW are usually calculated. This is interesting information and is usually discussed in the text of the article. Why are the polydispersity coefficients not presented and discussed in the article peptides?

R: Thank you for your comment. In particular, we are not used to working with the polydispersity coefficient, but it is a very interesting suggestion that we will take into account for our future work. In the case of the present article, it is not, in our opinion, vital that we include it in the analysis and discussion, especially since we have only 5 days to make and send all the revisions.

  1. The captions of figures 2, 3, 5, 9, 10 contain very long texts. Some of them can be transferred to the text of the article.

R: Thank you for your comment. We acknowledge that some figure captions are lengthy; however, we believe that the provided text is essential for ensuring clarity, particularly regarding the statistical analysis of the data presented in the graphs. Keeping this information within the figure captions allows readers to easily interpret the results without needing to refer back to the main text.

  1. The degree of purity of dimethyl sulfoxide should be indicated.

R: Thank you for your comment. The purity of dimethyl sulfoxide is indicated in the Materials and reagents section (Line 529).

Reviewer 3 Report

Comments and Suggestions for Authors

The manuscript  “Bioprospecting bioactive peptides in Halobatrachus didactylus body mucus: From in silico insights to essential in vitro validation” has great potential to discover new bioactive molecules that could have pharmaceutical, cosmetic, and food applications. Additionally, the use of computational approaches and experimental validation provides an effective framework to carry out the discovery of these compounds in an efficient and precise manner.

The manuscript presents a very good amount of information, well analyzed and discussed. However, I have some minor observations:

Line 15: (HdKTLR)……. What does Hd mean?

Line 19: aggregation at 344 mM and 700 mM ionic strengths at pH 7.0............... 344 mM millimoles of what?..........700 mM ionic strengths??????......... the ionic strength should not have concentration units. Please rewrite this idea, it's confusing.

Materials and methods should be placed after introduction

Line 94-95: replicating the conditions used to analyze the mucus peptide fraction……………. The procedure for analyzing the peptide fraction of the mucus has not been mentioned.

Line 96: specifically oligomers………… Why are they thought to be oligomers???

Line 100: 700 mM…. This is a measure of concentration, not ionic strength.

Line 101: 344 mM….. Same comment. Correct this throughout the manuscript.

Line 119: 700 mM..…………………….. This depends on the region, for example, in the Pacific Ocean the average salinity is 35 g/L, which corresponds to approximately 0.600 mM.

Line 258-260: 2.2.1.3. Mucus interaction and antioxidant capacity (ORAC assay)….mixing the peptides with the mucus to thoroughly investigate potential synergies……….. I don't see the need for this experiment. I recommend removing it.

Line 311: representing the mean ± standard deviation ......... This information is not necessary, I suggest deleting it.

Line 330: Abachi et al. ....... Please add the reference number

Author Response

The manuscript presents a very good amount of information, well analyzed and discussed. However, I have some minor observations:

Line 15: (HdKTLR)……. What does Hd mean?

R: Thank you for your observation. We would like to clarify that "Hd" represents the initials of the species name, with "H" for Halobatrachus and "d" for didactylus.

Line 19: aggregation at 344 mM and 700 mM ionic strengths at pH 7.0............... 344 mM millimoles of what?..........700 mM ionic strengths??????......... the ionic strength should not have concentration units. Please rewrite this idea, it's confusing.

R: Thank you for your comment. We would like to clarify that ionic strength refers to the measure of the total concentration of ions in a solution, calculated based on the molar concentration and charge of each ion present (Hay, Robert Walker. Reaction Mechanisms of Metal Complexes. Elsevier, 2000). You cans see also this definition in ScienceDirect of Elsevier: https://www.sciencedirect.com/topics/chemistry/ionic-strength#definition . Therefore, ionic strength is expressed in concentration units (mM), and we have maintained this notation accordingly in our article.

Materials and methods should be placed after introduction

R: Thank you for your comment. We would like to clarify that, in Marine Drugs, the standard manuscript structure follows the order: Introduction, Results, Discussion, Materials and Methods, and Conclusions (optional). This format aligns with the journal's guidelines, and we have structured our manuscript accordingly.

Line 94-95: replicating the conditions used to analyze the mucus peptide fraction……………. The procedure for analyzing the peptide fraction of the mucus has not been mentioned.

R: Thank you for your comment. We would like to clarify that the peptide fraction of the mucus was characterized in our previous study using the same SEC column, with a mobile phase consisting of a 344 mM phosphate buffer at pH 7.0. In this study, the same conditions were used to validate the molecular weights of the peptides. To ensure clarity for readers, we have explicitly stated in the text: “The HPSEC column was eluted with 344 mM phosphate buffer at pH 7.0, replicating the conditions used to analyze the mucus peptide fraction in our previous article [4]. This modification will be highlighted in red (Line 95).

Line 96: specifically oligomers………… Why are they thought to be oligomers???

R: Thank you for your comment. We would like to clarify our use of the term oligomers, which refers to molecules composed of repetitive units derived, either actually or conceptually, from smaller monomeric molecules. In this study, the designation of oligomers is based on the observed peptide aggregation phenomenon, which suggests the formation of structures consisting of multiple peptide units.

Line 100: 700 mM…. This is a measure of concentration, not ionic strength.

R: Thank you for your comment. As previously mentioned, ionic strength is expressed in concentration units (mM), following its standard definition based on the total concentration and charge of ions in solution. We have used this notation consistently throughout the manuscript in accordance with this definition.

Line 101: 344 mM…..Same comment. Correct this throughout the manuscript.

R: Thank you for your comment. As previously mentioned, ionic strength is expressed in concentration units (mM), following its standard definition based on the total concentration and charge of ions in solution. We have used this notation consistently throughout the manuscript in accordance with this definition.

Line 119: 700 mM..…………………….. This depends on the region, for example, in the Pacific Ocean the average salinity is 35 g/L, which corresponds to approximately 0.600 mM.

R: Thank you for your comment. We would like to clarify that the ionic strength of seawater at 700 mM is based on literature, specifically from Reference 10, which studies the Atlantic hagfish (Myxine glutinosa). The authors of that study used this value as a reference point. Furthermore, the fish species from which the peptides in our study were identified is native to the Portuguese coast in the Atlantic Ocean, further supporting the relevance of this reference.

Line 258-260: 2.2.1.3. Mucus interaction and antioxidant capacity (ORAC assay)….mixing the peptides with the mucus to thoroughly investigate potential synergies……….. I don't see the need for this experiment. I recommend removing it.

R: Thank you for your comment. However, we believe that this section is essential to our study, as it validates that mucus provides the ideal conditions for peptides to achieve high potency and enhanced antioxidant activity. Since mucus represents the natural environment of these peptides, its inclusion in the experiment is crucial for thoroughly assessing their biological potential.

Line 311: representing the mean ± standard deviation ......... This information is not necessary, I suggest deleting it.

R: Thank you for your comment. However, we consider this information is essential, as it indicates that the value of 577 ± 90 µg peptide/mL represents the mean and its standard deviation. This is crucial for demonstrating the reliability and reproducibility of the bioactivity determination.

Line 330: Abachi et al. ....... Please add the reference number

R: Thank you for your comment. We will include the reference number as suggested.